# New Insights into In Vivo Dopamine Physiology and Neurostimulation: A Fiber Photometry Study Highlighting the Impact of Medial Forebrain Bundle Deep Brain Stimulation on the Nucleus Accumbens

**DOI:** 10.3390/brainsci12081105

**Published:** 2022-08-19

**Authors:** Lidia Miguel Telega, Danesh Ashouri Vajari, Thomas Stieglitz, Volker A. Coenen, Máté D. Döbrössy

**Affiliations:** 1Laboratory of Stereotaxy and Interventional Neurosciences (SIN), Department of Stereotactic and Functional Neurosurgery, University Freiburg Medical Center, 79106 Freiburg im Breisgau, Germany; 2Department of Stereotactic and Functional Neurosurgery, University Freiburg Medical Center, 79106 Freiburg im Breisgau, Germany; 3Faculty of Biology, University of Freiburg, 79104 Freiburg im Breisgau, Germany; 4BrainLinks-BrainTools, IMBIT (Institute for Machine-Brain Interfacing Technology), University of Freiburg, 79110 Freiburg im Breisgau, Germany; 5Laboratory for Biomedical Microtechnology, Department of Microsystems Engineering (IMTEK), University of Freiburg, 79110 Freiburg im Breisgau, Germany; 6Bernstein Center Freiburg, University of Freiburg, 79104 Freiburg im Breisgau, Germany; 7Faculty of Medicine, University of Freiburg, 79110 Freiburg im Breisgau, Germany; 8Center for Basics in Neuromodulation, University of Freiburg, 79106 Freiburg im Breisgau, Germany

**Keywords:** deep brain stimulation, dopamine, fiber photometry, medial forebrain bundle, pulse width, major depressive disorder (MDD), rodents

## Abstract

New technologies, such as fiber photometry, can overcome long-standing methodological limitations and promote a better understanding of neuronal mechanisms. This study, for the first time, aimed at employing the newly available dopamine indicator (GRAB_DA2m_) in combination with this novel imaging technique. Here, we present a detailed methodological roadmap leading to longitudinal repetitive transmitter release monitoring in in vivo freely moving animals and provide proof-of-concept data. This novel approach enables a fresh look at dopamine release patterns in the nucleus accumbens, following the medial forebrain bundle (mfb) DBS in a rodent model. Our results suggest reliable readouts of dopamine levels over at least 14 days of DBS-induced photometric measurements. We show that mfb-DBS can elicit an increased dopamine response during stimulation (5 s and 20 s DBS) compared to its baseline dopamine activity state, reaching its maximum peak amplitude in about 1 s and then recovering back after stimulation. The effect of different DBS pulse widths (PWs) also suggests a potential differential effect on this neurotransmitter response, but future studies would need to verify this. Using the described approach, we aim to gain insights into the differences between pathological and healthy models and to elucidate more exhaustively the mechanisms under which DBS exerts its therapeutic action.

## 1. Introduction

Brain stimulation strategies are becoming the key treatment options in a wide range of neuropsychiatric diseases. In clinical trials with treatment-resistant depression (TRD) patients, the chronic deep brain stimulation (DBS) of the superolateral branch of the medial forebrain bundle (slMFB) has been shown to rapidly and chronically ameliorate the depressive symptoms [1,2,3,4,5,6,7]. Pre-clinical research is being used to study the possible modes of action of DBS in rodent models of depression [8,9,10,11,12,13,14,15].

One hypothesis concerning the positive clinical and pre-clinical effects of mfb DBS is that stimulation can mediate anti- and orthodromic activity in key regions of the dysfunctional reward pathway: the ventral tegmental area (VTA), the nucleus accumbens (NAc) and the medial prefrontal cortex (mPFC). According to this idea, mfb DBS first modulates the glutamatergic fibers connecting mPFC with VTA and, subsequently, promotes an indirect action on NAc dopamine dynamics through the VTA-NAc neuronal dopaminergic activation [7,16].

Depression is a multifactorial disorder, and there are numerous mechanisms associated with its pathophysiology. The current paper focuses on dopamine, as this transmitter plays a pivotal role in motivation, reinforcement learning, reward-related processes [17,18] and is seminal in anhedonic states (loss of pleasure) characterized in the major depressive disorder (MDD) [19,20].

Several questions remain concerning DBS stimulation parameters, even though the typical clinically used chronic high-frequency stimulation parameters (frequency: 130 Hz, pulse width: 60 µs) appears to be efficacious in principle. The physical and anatomical properties of neuronal fibers associated with the pathology are key determinants in the excitability of the fibers, which, in turn, should dictate the stimulation parameters applied. Attending to these factors and improving our understanding of the parameters are likely to result in differential and better treatment outcomes [21].

Dopaminergic and glutamatergic fibers differ in their physical and anatomical features (non-myelinated vs. myelinated, smaller vs. larger diameters, respectively) (Yeomans, 1989), thus, responding differently to the amount of current injected and creating different release and activation patterns in the regions they feed into or come from [22,23]. In the past, studies looked at the self-stimulating effects of mfb DBS parameters on the refectory periods of neurons [24,25]. More recently, using electrophysiology or voltammetry, studies looked at how stimulation parameters affect the release patters in different neuronal subtypes—for example, in dopaminergic neurons [26,27]. A recent study from our group by Ashouri and colleagues using the fast-scanning cyclic voltammetry (FSCV) applied the principle of chronaxie (stimulation duration vs. fiber electrical response relation) and showed that different DBS PWs can have a differential effect on the dopamine release pattern in the NAc between healthy and rodent models of depression [28]. More refined research is needed to investigate the impact different DBS parameters may have on different targeted structures and neurotransmitters.

Currently, the main experimental options to measure in vivo DBS evoked neurotransmitter release are dialysis or voltammetry. However, neither of the two methods is adapted to perform repetitive and longitudinal measurements looking at the dynamics of neurotransmitter release over time in rodent models of disease. To facilitate a chronic and stable monitoring of neurotransmitter projections in the brain, we tested fiber photometry (FP), which offers a more appropriate approach to answer our experimental questions, and comprehensive descriptions of this technique can be read elsewhere [29,30,31]. FP can be used to monitor fluorescent sensors associated with general or subtype selective neural activity in specific brain regions. In this method, an optic fiber, implanted targeting a specific brain structure, is used to deliver the excitation light and to absorb the emitted fluorescence from the biosensor, which is then passed on to the photo detector and then to the amplifier for further signal processing. Molecules of interest are tagged using genetically encoded fluorescent indicators (GEIs) and can be quantified via exciting the GEIs to a higher energy state and absorbing the emitted fluorescence. The GEIs are injected in the targeted brain structures by means of viral plasmids, and recordings take place with the optic fibers using LED excitation. Those GEIs, in the specific case of GRAB_DA2m_, employ the naturally occurring dopamine D2 receptors expressing a green fluorescent protein (GFP) in their third intracellular loop. The moment dopamine binds the genetically encoded dopamine indicator receptors, the whole complex, by means of a ligand-stabilized conformational change, increases the levels of emitted green fluorescent light, which can then be quantified and compared to the baseline levels of emission.

The current work aimed to establish the protocol and offer a proof of principle of using fiber photometry (i.) to investigate the relationship between mfb DBS parameters and dopamine release in the NAc and (ii.) to do so in a longitudinal fashion in a small number of freely moving healthy rodents. The fast development and refinement of biosensors available for fiber photometry [32,33] has made it now possible to better investigate the real-time acute and chronic effects on selective neurotransmitter release of mfb DBS and thus better understand the mechanisms behind its depression-alleviating effects.

## 2. Materials and Methods

### 2.1. Animals—Experimental Design

Three Long–Evans rats (1 female, 2 males, in-house bred) weighing ~250–300 g at time of surgery were single housed after surgical implantations. Water and food pellets were provided *ad libitum*. Light cycle was scheduled from 7 am to 7 pm, and temperature and humidity were constantly monitored (22 ± 1 °C and 50–60%, respectively). Rats were habituated to the head implant for four days before the start of the stimulation/recordings. Recordings started four weeks after surgery to allow for viral expression. All procedures were performed according to the regional ethics committee Regierungspräsidium Freiburg (TierSchG), following the EU-directive 2010/63/EU.

The experimental design (Figure 1A) consisted of a period of recordings (once/week) over two weeks during which different PWs were employed sequentially in every session (100, 250 and 350 µs). A single session consisted of 5 min baseline recording and then 5 s (wk1) or 20 s (wk2) of 130 Hz square biphasic DBS pulses with one of the three specific PWs, followed by 5 min post-stimulation recording. Stimulation amplitudes were titrated for each animal by identifying the minimum current inducing a SEEKING behavioral response (100–300 µA). The SEEKING phenotype can be described as stimulation evoked, rapid and transient (closely associated with the onset and termination of the mfb DBS) increase in the explorative, searching and rearing behavior (for description, see Ref [20]). The whole stimulation PW pattern of three different PWs (block 1, 2, 3) was repeated 3 times, with a 30 min reco PW ery time in between repetitions.

### 2.2. Surgery (Viral Construct, Optic Fiber, DBS Electrode)

Rats were anesthetized with 4–5% isoflurane, placed in the stereotactic frame and maintained at the 1–2.5% level with continuous oxygen flow (1 L/min). During surgery, rats received ~1 mL of 5% glucose twice (pre- and post-implantation), and at the end, once anesthesia was stopped, 0.05 mg/kg Buprenorphin was injected subcutaneously. Viral injections of the dopamine sensor GRAB_DA2m_ (AAV9-hsyn-DA2m(DA4.4); titer: 4.37E + 13; WZ Biosciences Inc, Columbia, MD, USA, [32]) were infused using a 2000 nL Hamilton^®^ (Hamilton Company. Reno, NV, USA) syringe into two different coordinates of the left hemisphere (targeting the NAc shell) to allow a good spread and covering of the targeted region: (1) AP: +1.6, ML: +1.0, DV: −7.0; (2) AP: +1.2, ML: +1.0, DV: −7.0. The rate of injection was set to 100 nL/min, and the total volume injected per coordinate was 200 nL. After each injection, 2 min of waiting time were employed before removing the injection needle in order for the viral solution to absorb in the tissue and avoid spread out when removing the needle. Optic fiber (material: borosilicate, receptacle: metal ferrule MF2.5, numerical aperture: 0.66, 400 µm ⌀, Doric Lenses^®^ (Doric Lenses Inc., Quebec City, QC, Canada) was implanted in the NAc (AP: +1.4, ML: +1.0, DV: −6.8). DBS electrode (double-helix custom-made perfluoralkoxy-polymere (PFA) coated wire 90% Platinum−10% Iridium, bare wire 127 µm ⌀, coated single wire 200 µm ⌀, Science Products GmbH, Hofheim am Taunus, Germany) was implanted in the mfb (AP: −2.8, ML: +0.7, DV: −7.7). All targeted coordinates are expressed relative to bregma after “flattened the skull”—same dorso-ventral coordinates for bregma and lambda. Two anchor mini-screws 1.2 mm ⌀ (Bürklin, Oberhaching, Germany) were screwed into the skull near by the implants and later fixated employing bone cement (“Palacos”, Heraeus, Hanau, Germany) to achieve a rigid implant in the skull. Electrode wires were connected to a 6-pin female socket (3 pins with 2 rows) cut from a socket connector, 50-pin, RM 1.27 mm, straight, 10005965, BRL 250 type (Fischer Elektronik, Bürklin, Oberhaching, Germany).

### 2.3. Fiber Photometry (FP) Recordings—Data Analysis

In vivo recordings were performed in an empty experimental box within the experimental area. Patch cord power was measured each time before the recordings. The animal was connected to the FP patch cord using a mating sleeve (Doric Lenses^®^ Inc., Quebec City, QC, Canada), and the DBS cable male connector was joined with the implanted female header connector. Fluorescence minicube, LEDs, LED drivers, fluorescence detector (Doric Lenses Inc., Quebec City, QC, Canada) and Synapse Suite Version 94 software and RZ5 BioAmp (Tucker-Davis Technologies, Alachua, FL, USA) were used to record the photometric signals. LED green fluorescence protein (GFP)-sensitive and isosbestic (GFP-non-sensitive, used to remove motion artifact and fluorescence bleaching) excitation signals were 465 nm and 405 nm, respectively. The signal was digitized at 6 kHz. The transistor–transistor logic (TTL) DBS pulses generated by a custom-made stimulator were sent to the RZ5 BioAmp processor and co-registered with the fluorescent signals.

Data were analyzed using pMAT v1.2 ([34]; The Barker Lab, Philadelphia, PA, USA) and custom-made scripts in MATLAB R2019b (The MathWorks Inc., Natick, MA, USA). Raw GFP signal (465 nm, *F*_465_) from GRAB_DA2m_ dopamine sensor activity was fitted using the isosbestic signal (405 nm, *F*_405_) in order to obtain the corresponding dopamine fluorescence $\left(\frac{\Delta F}{F}\right)$ free of the movement artifacts and any other biological processes not related to dopamine (Equation (1), [34]).
(1)$$\frac{\Delta F}{F}\;=\;\frac{F_{465}-F_{405\left(scaled\;control\;channel\right)}\;}{F_{405\left(scaled\;control\;channel\right)}\;}$$

The scaling of the control channel was performed using the polyfit MATLAB integrated function. The final co-registration and normalization (*z*-*score* (Equation (2), [34]), number of standard deviations from the baseline) of the DBS events (Event) was carried out using 5 s baseline sampling window taken before the DBS event and using a 100-bin constant in order to extract: (a) the area under curve (AUC) (portion of curve falling within the defined window using trapezoidal method MATLAB trapz), (b) maximum of the fluorescence signal (peak occurring under each time window) and (c) the times at which they occurred, in order to account for the response dynamics of mfb DBS and possibly be able to reveal and explore further trans synaptic activation of midbrain dopamine neurons.
(2)$$\frac{\Delta F}{F}\;z-score\;\left(i\right)=\frac{\left[\frac{\Delta F}{F}\;DBS\;Event\;\left(i\right)-median\left(\frac{\Delta F}{F}\;baseline\right)\right]}{median\;absolut\;deviation\;\left(MAD\right)\;of\;baseline}$$

The individual *z*-*scores* were obtained for the following time periods in reference to the DBS event: (a) baseline: −5 to 0 s, (b) during DBS: 0 to 5 s (week 1) or 0 to 20 s (week 2) and (c) 5 or 20 s post-stimulation. In the 20 s DBS case, the 20 s post-stimulation period was divided into 5 s intervals to visualize the time effects in more detail. The visualization of data was performed using GraphPad Prism 9.0.0^®^ (GraphPad Software, San Diego, CA, USA) and MATLAB R2019b^®^(The MathWorks, Natick, MA, USA).

### 2.4. Immunohistochemistry

Rats were anesthetized i.p. with 100 mg/kg ketamine (10% Medistar GmbH, Ascheberg, Germany) combined with 10 mg/kg xylazine (2% Rompun^®^, Bayer, Leverkusen, Germany) and then perfused with saline and 4% PFA. Brains were kept in 4% PFA overnight. Later, they were replaced with a 30% sucrose solution. Brains were sliced in a microtome into 40 µm sections and embedded in a tissue-freezing medium. Posteriorly, the electrode, optic fiber placement and virus expression were verified by staining against GFP and Tyrosine Hydroxylase (TH) (mouse (ms) anti-GFP A11120 Invitrogen (1:500) and rabbit (rb) anti-TH AB152 Merck (1:800) antibodies, respectively), followed by incubation with secondary antibodies goat (Gt) anti-ms 488 A11001 Life Tech. (1:200) and Gt anti-rb 568 A11011 Life Tech. (1:200). In both cases, the antibody solutions were diluted in 1% Bovine Serum Albumin (BSA) and 0.3% PBS-Tx. Slices were scanned with a fluorescence microscope (AxioImager2 20x objective Zeiss, Jena, Germany).

### 2.5. Statistical Analysis

All results are presented as mean with standard error of the mean (SEM) for the bar plots. Data were checked for normality using the Shapiro–Wilk test. A non-parametric Friedman ANOVA test was used to determine statistical significance among groups when normality was not fulfilled. Normally distributed data were analyzed for each week of recording (5 s or 20 s DBS) using two-way repeated measures (RM) ANOVA with Greenhouse–Geisser correction for adjusting the lack of sphericity, and when applied, post hoc Bonferroni correction (factors: PW, time interval and their interaction). The time of maxima was assessed for statistical significance using one-way RM ANOVA for each week of recording. All statistical analysis was performed using GraphPad Prism 9.0.0^®^ and OriginPro 2019b^®^ (OriginLab, Northampton, MA, USA). Statistical significance was set as: *p* < 0.05 *, *p* < 0.01 **, *p* < 0.001 ***, *p* < 0.0001 ****.

## 3. Results

The histological verifications confirmed the correct placement of the electrodes, optic fibers and viral expression in the desired target regions (mfb, NAc, Figure 1B, left and right images, respectively). The intensity of the fluorescence indicator signal after GFP tissue staining was in the range of 312.8 ± 43.0 a.u. (arbitrary units) in an area of 1.93 × 10^6^ ± 0.09 × 10^6^ µm^2^, showing a well-distributed and highly intense signal within the desired target area to perform the FP recordings. The fluorescent levels in the opposite hemisphere (not tagged with the virus): 180.7 ± 17.1 a.u. in the delimited mean area of 2.00 × 10^6^ ± 0.11 × 10^6^ µm^2^. The power at the tip of the optic fiber measured before each recording was 50.5 ± 1.2 µW, which remained stable across the sessions. The dopamine response obtained after each single stimulation trial and DBS parameter employed generated an increased dopamine response in the NAc with respect to its baseline activity for both DBS stimulation conditions, which decreased along the stimulation time (5 s, Figure 2A and 20 s, Figure 2B). Baseline activity dynamic scores were similar for all PWs and DBS sessions (AUC for all PWs–5 s DBS: 35.96 ± 33.65 a.u.; AUC for 20 s DBS: 10.55 ± 34.35 a.u.; Figure 2C,D). Once DBS was applied, these values attained a remarkable increase for all PWs, which were higher for the 20 s DBS than for the 5 s DBS (mean AUC *z*-*score* for all PWs and trials during stimulation: 323.5 ± 65.2 and 409.1 ± 97.2 a.u., 5 s and 20 s, respectively). However, within the 5 s DBS, 350 µs PW was able to induce a slightly higher dopamine response than the shorter PWs (Figure 2C,E). After the DBS was off, the dopamine photometric readouts returned close to their previous baseline level or below it, the latest more predominantly for all PWs in the 20 s DBS case (Figure 2D). Overall, none of the changes, either for the 5 s or the 20 s DBS, attained statistical significance after performing the two-way RM ANOVA (5 s DBS–time interval factor: F(1.05, 2.12) = 7.1, n.s. *p* = 0.11; PW: F(1.28, 2.56) = 6.89; n.s. *p* = 0.92; interaction: F(1.02, 2.04) = 0.76, n.s. *p* = 0.48; 20 s DBS–time interval factor: F(1.05, 2.10) = 2.20, n.s. *p* = 0.27; PW: F(1.02, 2.03) = 0.11, n.s. *p* = 0.77; interaction: F(1.02, 2.04) = 0.28, n.s. *p* = 0.65), thus showing that the separated range of PWs used was not able to register predominant and differential variations in the dopamine response using photometric readouts.

The peak amplitudes between the different DBS stimulation times did not vary much; the maximum mean *z*-*score* for 5 s DBS and all PWs was 20.2 ± 2.3 a.u. and for 20 s DBS: 21.3 ± 2.7 a.u. Statistical analysis only revealed a time interval significance after two-way RM ANOVA for the 5 s DBS with no post hoc Bonferroni significant variations among the groups (time interval: F (1.22,2.45) = 26.82, * *p* = 0.02, PW: F (1.16,2.33) = 5.5, n.s. *p* = 0.13; interaction: F (1.09,2.17) = 0.53, n.s. *p* = 0.55; Figure 2E). However, the 20 s DBS, did not show significant changes between any of the time intervals (Friedman ANOVA: ꭕ^2^_F_ (1) = 3, *p* > 0.05; Figure 2F). Interestingly, the time at which these peak amplitudes were achieved did not vary for any of the cases, DBS duration or parameters used (5 s DBS–100 µs: 0.66 ± 0.05 s, 250 µs: 0.66 ± 0.02 s, 350 µs: 1.24 ± 0.46 s; 20 s DBS–100 µs: 0.70 ± 0.04 s, 250 µs: 0.65 ± 0.06 s, 350 µs: 0.73 ± 0.05 s; Figure 2G), except for the outlier presented in the 5 s DBS–350 µs case. One-way RM ANOVA did not reveal statistically significant variations for any of the PWs in any of the DBS duration conditions (F (1.2,2.5) = 0.72, n.s. *p* = 0.50), thus showing very similar time dynamics on the dopamine response, independent of the DBS parameter employed.

## 4. Discussion

The A10 midbrain mesocorticolimbic dopaminergic projection originates in the VTA and projects via the medial forebrain bundle (mfb) to the NAC and dorsolateral prefrontal cortex [35,36]. This pathway, referred to as the SEEKING system in the *affective neuroscience* literature, is considered to act as the neural substrate for positive emotional and euphoric behaviors supporting exploration and controlling appetitive learning [37,38]. The mesocorticolimbic dopaminergic pathway is also associated with complex emotion regulations affected in clinical depression, such as “wanting”, “desiring”, “anticipating” and “hoping”; in experimental research, these functions translate into motivation, anhedonia or reward-orientated behavior [39,40,41].

The current study, for the first time, used the dopamine indicator (GRAB_DA2m_) in combination with fiber photometry to investigate the physiology under the action of mfb DBS and to assess dopamine release longitudinally at repeated time points. This novel technique overcomes many of the the standard techniques’ limitations and opens up a path toward further probing of acute and chronic DBS mechanisms. In both of these methods, there are issues concerning detection sensitivity and selectivity of neurotransmitters, molecular inaccuracy or, for instance, the impossibility to target electrochemically inert s ubstances, such as glutamate, or to conduct chronic (e.g., 2–6 weeks) studies. So far, a single fiber photometric study has explored the mediated DBS effects integrating photometric readouts, although stimulating and recording directly in the NAc in the context of a model of eating disorders [42], hence also showing the novelty of this technique to elucidate DBS mechanisms in an electrical artifact free recording platform.

In relation to our previous work [28], the current approach overcomes the limitations of restricting the recordings to acute readouts in anesthetized animals, as we were able to achieve repeated recordings over time in freely moving animals. We demonstrated that the proposed FP medium affinity dopamine biosensor (GRAB_DA2m_), D2-receptor-based, with increased dynamic range in comparison to previous-generation sensors (GRAB_DA1m_), was able to detect reliably and in a consistent way downstream dopamine changes induced by direct electrical modulation of the mfb fibers in a highly spatio-temporal resolved fashion. The approach allows for consistent longitudinal tracking of the induced and aggregated focal dopamine neuronal population activity without losing optical resolved capacity in time (photo-bleaching effects). This confirms its potential use in chronic “depressive-like” rodent models, where repetitive longitudinal monitoring—in combination with behavioral testing—is essential. It also must be noted that the sensor here employed is dopamine D2 receptor based, therefore, accounting for different processes as to the subtype of indicators based on dopamine D1 receptors like dLight [43]. In this way, we were able to discern between the physiological binding of dopamine among one of the different receptor subtypes, the one more involved in motivational-related processes, and therefore DBS mediated “anti-depressive-like” states [44,45]. From the molecular and technical point of view, the D2-receptor-based indicator used accounts for higher signal to noise ratio than the dLight sensors, with only slower off-kinetics [32], which slightly hinder the visualization of some faster and more fluctuating dynamical processes by a summative process. However, from the practical point of view, the employed sensor is suitable for the DBS modulation processes used and within the stimulation times employed with the presented on-kinetics of the sensor.

The acute activation of the dopamine system following mfb DBS has already been shown in healthy and depressive-phenotype animal models [26,28,46], in self-stimulation studies [22,27,47] or in studies using drugs of abuse [47]. Accordingly, our findings also suggest the direct potentiation of dopamine release in the NAc during mfb DBS. However, there are also various studies showing contradictory results, related more specifically to behavioral outcomes [8,23], which provide the need to further corroborate the effects of DBS on dopamine transmission.

In our analysis of the PW findings, we did not observe differential significance across this parameter at either DBS duration. A differential tendency was noted under the 5 s DBS 350 µs PW condition. However, longer DBS intervals (20 s DBS) showed to have a longer and more sustained effect on the dopamine dynamics and recovery back to a more negative baseline state compared to the shorter 5 s stimulation, suggesting different dynamics on the release patterns. This observation is likely due to a different pathway modulation, type and/or amount of fibers recruited or activation time, and it is probably related to the specific time-dynamics characteristics of GRAB_DA2m_.

The slightly delayed peak amplitudes could be related to the hypothesis of an indirect or trans-synaptic activation of dopaminergic fibers [7,16]. The delayed peak amplitudes are probably independent of the PW within the range used here (100–350 µs), since higher PWs could potentially have a different dopamine time response, as studies have shown longer PWs (>500 µs) to have stronger effects on the refractory periods of neurons [24,25]. In our case, we cannot rule out an implication of different fiber-type activation through the use of different PWs nor judge its effectiveness, but we validate the effects on dopamine release, as it is not the scope of this study. However, further evidence would be needed to assert both assumptions.

The variabilities (outliers) found within the study (inter-individual differences across animals) could be due to the small sample size and the methodological related inaccuracies during the stereotactic surgeries. For example, differences in the precision of the optic fiber implantation, intractable control of the amount of cells tagged with the dopamine indicator and mfb DBS electrode implantation position within the mfb might result in the activation of different amounts of fiber subtypes (dopamine thin unmyelinated vs. glutamatergic thick myelinated fibers) [22], also accounting for a functional heterogeneity by their target structures [48] and, therefore, in the amount of quantified dopamine response.

The study has several limitations, one of which is the sample number. To address these issues in the future, we will use larger cohorts to increase the power and the reproducibility of the study. Moreover, we will extend the measurements to disease and healthy models, to male and female animals (with no bias), and measurements will be taken during behavioral tasks. Future studies will also investigate other DBS parameters and stimulate separately either a certain proportion of mfb fibers (dopaminergic vs. glutamatergic) or certain pathways within the same bundle targeting different subregions of NAc or other MDD related targets, such as mPFC. To address a technique-based limitation concerning the relatively low (second scale) temporal resolution, we will combine FP and electrophysiological recordings to provide the sub-second population measurements through electrophysiology and at the same time have cell-specific, chronic and also artifact-free recording platform offered by FP. Better understanding of how parameters might differentially affect and regulate dopamine or glutamate release could improve future clinical application of DBS by implementing context-dependent selective neurotransmitter-based adaptive DBS strategies.

## Figures and Tables

**Figure 1 brainsci-12-01105-f001:**
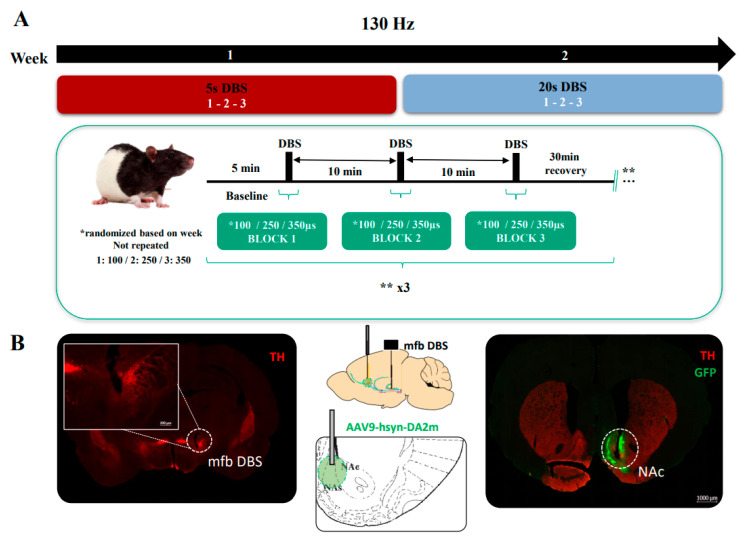
(**A**) Experimental design. Long–Evans rats received, per session, 5 min baseline recording followed by 5 s (week 1) or 20 s (week 2) 130 Hz DBS with a titrated mean amplitude of 200 µA, with 10 min inter-stimulation. The whole design was repeated 3 times for each different DBS parameter (100/250/300 µs PWs) with a 30 min wash-out period in between repetitions. (**B**) Schematics of the surgical target regions (middle images). Histological verification of the mfb DBS electrode (left image), optic fiber implantation and viral injection sites in the NAc (right image). Scale: 1000 µm. Magnified scale: 200 µm.

**Figure 2 brainsci-12-01105-f002:**
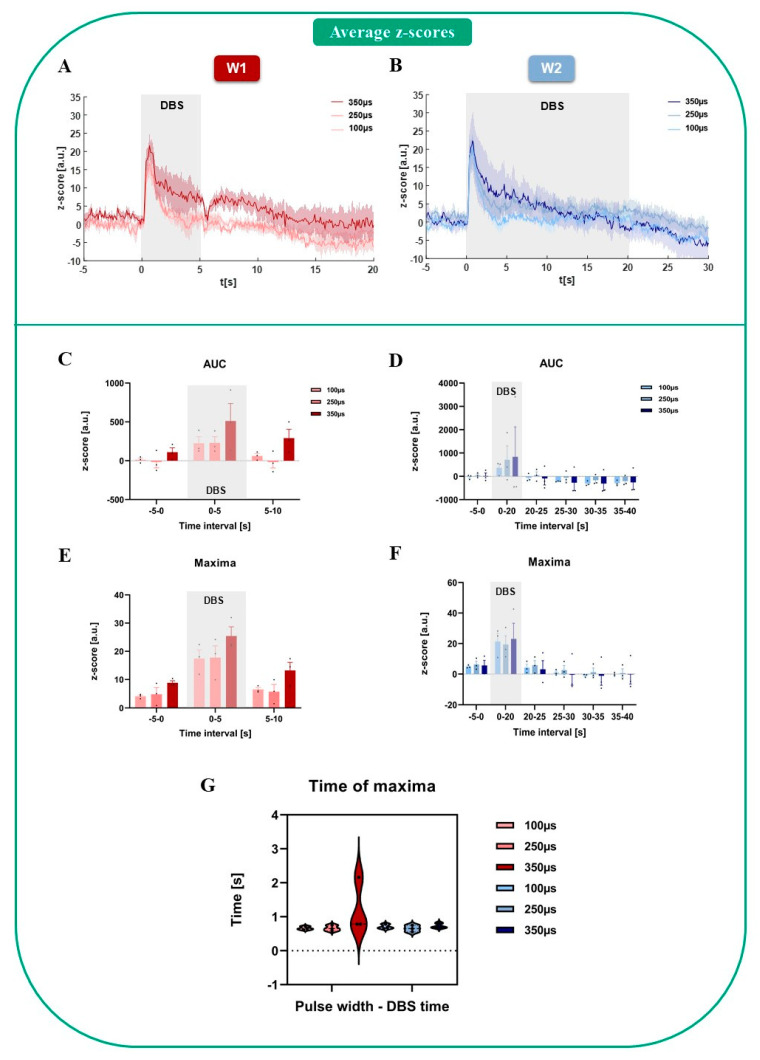
(**A**) Average dopamine profiles (*n* = 3) for 5 s DBS (week 1 (W1)) and all PWs (100, 250 and 350 µs). (**B**) Average dopamine profiles (*n* = 3) for the 20 s DBS (week 2 (W2)) and all PWs (100, 250 and 350 µs). (**C**) Area under the curve (AUC) for the baseline period (−5–0 s), 5 s stimulation (0–5 s) and 5 s post-stimulation (5–10 s) for all PWs, *n* = 3. (**D**) Area under the curve (AUC) for the baseline period (−5–0 s), 20 s stimulation (0–20 s), 5 s post-stimulation (20–25 s), 10 s post-stimulation (25–30 s), 15 s post-stimulation (30–35 s) and 20 s post-stimulation (35–40 s) for all PWs, *n* = 3. Overall maximum increase in dopamine response during the first second of stimulation and progressive return back to baseline and below baseline level. (**E**) Maximum amplitude of the dopamine fluorescence signal achieved for all time intervals (pre-stimulation, 5 s stimulation and post-stimulation). Two-way RM ANOVA significant time effects found (*p* = 0.0224, *p* < 0.05). (**F**) Maximum amplitude of the dopamine fluorescence signal achieved for all time intervals (pre-stimulation, 20 s stimulation and post-stimulation). (**G**) Time at which maximum dopamine fluorescence signal was achieved after start of stimulation. Similar times achieved for all PWs and conditions.

## Data Availability

All original data leading to this paper are stored on a computer located at Laboratory of Stereotaxy and Interventional Neurosciences (Department of Stereotactic and Functional Neurosurgery, University Freiburg Medical Center) in the office of L.M.T. All raw data can be made available upon request.

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
