# Peer review of "New Insights into In Vivo Dopamine Physiology and Neurostimulation: A Fiber Photometry Study Highlighting the Impact of Medial Forebrain Bundle Deep Brain Stimulation on the Nucleus Accumbens"

_brainsci, 2022, doi:10.3390/brainsci12081105_

Round 1

Reviewer 1 Report

In this study, the authors assessed the effects of DBS using two methods, dopamine indicators and fiber photometry, in which authors made a good effort and had a well-focused discussion.

 I have a few questions as follows,

1)      I'm a little confused about the experimental design. For example, 5s-DBS was performed in the first week (once), so how much time did this task take in total? On what day of the week? Please clarify the timeline.

2)      Following the question1, different pulse widths in the task are performed in sequence. Will different pulse widths affect each other? Why not use a single pulse width?

3)      For surgery (virus, optic fiber), is it generally performed on one side (left) or both sides? Is there any difference.

4)      Very nice discussion.

Reviewer 2 Report

Major comments:

1.     “360.7 ± 51.3 a.u.” fluorescence intensity alone does not make sense. Just refer to the image or make a quantification and comparison with the same area in the opposite hemisphere.

2.     The authors likely used Geisser-Greenhouse correction for ANOVA and should state that in the methods.

3.     Discussion about the significance of the DA receptor subunit does not make sense, as these GPCR-based sensors are inert (at least mostly) in terms of the intracellular cascades, their localization profile is unlikely to follow the physiological one, etc, thus the DA1 or DA2 based sensors both sense the same DA…

4.     Related to the previous comment, authors should better describe the operating principle (more technically). Also, the authors should state some limitations in the discussion.

5.     When describing results, the authors should clearly state if their observations are statistically verified and state the specific test in parenthesis, even when not significant. When just suggesting some visible trend, please state it clearly (e.g: … suggests, seems to etc… although the difference was not statistically significant)

6.     The sample size is very low (N = 3) and gender biased. These limitations must be clearly stated, and the authors must provide statistical reasoning for this choice (or technical reasons for exclusion of samples). In addition, with such a low N, the statistical significance likely suffers from poor power. Hence, the authors must provide an estimation of the statistical power together with p.

Minor comments:

1.     rephrase the wrong sentence “first modulates the efferent glutamatergic fibers connecting mPFC and VTA and subsequently, promoting an indirect action on NAc dopamine dynamics through the VTA-NAc projections [7,16].”

2.     Replace biology in “The biology of depression is multifactorial”

3.     “The current line of research focuses on dopamine” -> one of the most investigated…

4.     “efferent vs afferent” makes no sense here

5.     Check spelling in equations (z-scores)

6.     All text should be revised for improving English

Round 2

Reviewer 2 Report

All issues have been addressed